# Azole-Resistance Development; How the *Aspergillus fumigatus* Lifecycle Defines the Potential for Adaptation

**DOI:** 10.3390/jof7080599

**Published:** 2021-07-24

**Authors:** Jianhua Zhang, Alfons J. M. Debets, Paul E. Verweij, Eveline Snelders

**Affiliations:** 1Laboratory of Genetics, Wageningen University & Research, 6700 Wageningen, The Netherlands; jianhua.zhang@wur.nl (J.Z.); fons.debets@wur.nl (A.J.M.D.); 2Department of Medical Microbiology, Radboudumc-CWZ Center of Expertise for Mycology, Radboud University Medical Center, 6500 Nijmegen, The Netherlands; paul.verweij@radboudumc.nl

**Keywords:** *Aspergillus fumigatus*, azole resistance, lifecycle, adaptation, recombination, mutation

## Abstract

In order to successfully infect or colonize human hosts or survive changing environments, *Aspergillus fumigatus* needs to adapt through genetic changes or phenotypic plasticity. The genomic changes are based on the capacity of the fungus to produce genetic variation, followed by selection of the genotypes that are most fit to the new environment. Much scientific work has focused on the metabolic plasticity, biofilm formation or the particular genetic changes themselves leading to adaptation, such as antifungal resistance in the host. Recent scientific work has shown advances made in understanding the natural relevance of parasex and how both the asexual and sexual reproduction can lead to tandem repeat elongation in the target gene of the azoles: the *cyp*51A gene. In this review, we will explain how the fungus can generate genetic variation that can lead to adaptation. We will discuss recent advances that have been made in the understanding of the lifecycle of *A. fumigatus* to explain the differences observed in speed and type of mutations that are generated under different environments and how this can facilitate adaptation, such as azole-resistance selection.

## 1. Introduction

*Aspergillus fumigatus* is a filamentous fungus that can cause human diseases ranging from allergic bronchopulmonary aspergillosis to chronic pulmonary aspergillosis or even life-threatening acute invasive aspergillosis (IA). The interactions between *A. fumigatus* and the host environment are dynamic and complex. *A. fumigatus* is ubiquitous in our environment, and asexual spores can be dispersed over wide geographic distances by air currents. *A. fumigatus* is a saprotrophic fungus that is found in soil and grows on decaying organic matter, and it has an important role in carbon and nitrogen recycling. Humans are estimated to inhale at least 100 conidia each day, and due to their small size, a fraction of inhaled conidia will reach into the alveoli of the lungs [1]. In immune-competent individuals, airway epithelial cells and resident alveolar macrophages remove inhaled conidia; however, in individuals who are unable to clear these conidia, germination could occur, subsequently leading to *Aspergillus* disease [2]. Among the infections caused by *Aspergillus* species, *A. fumigatus* is the leading etiological agent in most geographic regions. IA may occur in patients with specific immune defects, such as neutropenia, and infection is associated with substantial morbidity and mortality. Recently, cases of IA have been increasingly reported in patients with severe viral pneumonia, including influenza and coronavirus disease 2019 (COVID-19) [3,4,5,6]. Despite the considerable advances made over the past decades in the management of fungal infection, such as diagnostics and antifungal therapy [7], the management of IA has become increasingly complex due to various reasons, including acquired antifungal resistance [8]. Resistance to voriconazole was shown to be associated with excess mortality in voriconazole-treated IA patients, compared to patients with voriconazole-susceptible IA [9]. Furthermore, *A. fumigatus* is capable of persisting in the lungs of patients with structural lung diseases, such as COPD and cystic fibrosis (CF).

In order to successfully infect or colonize human hosts, *A. fumigatus* may initially establish lung colonization due to its physiological versatility and can subsequently adapt, through genetic changes, to the human lung environment and stressors, such as antifungal agents. Much scientific work has focused on the metabolic plasticity, biofilm formation [2] or the particular genetic changes themselves leading to adaptation, such as antifungal resistance in the host [10,11]. In this study, we review what is currently known of the lifecycle of *A. fumigatus* in relation to various environments. We focus on the various aspects of the lifecycle and their relevance for contributing to adaptive genetic variation by mutation and recombination under the different environmental conditions. Furthermore, we discuss the implications of genetic variation for azole resistance, a selection factor found both in human medicine and the environment.

## 2. *A. fumigatus* Lifecycle

### 2.1. Asexual Lifecycle

An *A. fumigatus* colony is initiated from a single spore or hypha on a suitable substrate. Within four to six hours, conidia can develop short hyphae, known as germ tubes, that, by mitotic division and branching, form an extended mycelial network. Within a few days, asexual spores, also called conidia, are formed on specialized hyphal structures, called conidiophores. The spore head of a conidiophore can produce up to 10⁴ asexual spores, and the dispersal of asexual *A. fumigatus* conidia in the environment is highly efficient, as these conidia are very hydrophobic compared to other Aspergillus species and are very efficiently spread by air [2,12]. Conidia of *A. fumigatus* do not germinate or swell in distilled water but germinate under nutrient- and oxygen-rich conditions [7]. Germ tubes or tubular hyphae grow in a polar fashion by apical extension and branching to form a network of interconnected cells, known as a mycelium. A colony consists of septate multinucleated cells that grow in a radial shape [13]. The structure of the cell wall of the mycelial or vegetative form of filamentous fungi is different from the cell wall structures of conidia and conidiophores, of which the latter functions for survival and dissemination (Figure 1). The fact that *A. fumigatus* has a high mycelial growth rate and rapidly produces abundant airborne conidia promotes the colonization of multiple environmental niches [2].

### 2.2. Parasexual Lifecycle

Parasexual recombination in fungi, i.e., recombination outside the sexual cycle, has been suggested as an alternative to sex in nature for generating diversity [14]. The parasexual cycle, was initially described to have the following elements: (1) heterokaryon formation following anastomosis of hyphae from vegetatively compatible but genetically different colonies; (2) heterozygous diploid formation by nuclear fusion of genetically unlike nuclei, multiplication of the diploid nuclei in the heterokaryon and segregation into diploid spores that, after dispersal, may establish diploid colonies; (3) recombination during mitotic divisions of diploid nuclei by crossing over; (4) nondisjunction, leading to haploidization. First observed in *A. nidulans*, parasex has later been discovered in many other fungi, including *A. fumigatus*, [15], and was considered to be especially relevant for fungi that are predominantly or completely asexual. However, when analyzing natural isolates of various fungi, it was found that most are not capable of forming stable heterokaryons, a prerequisite for parasex. Heterokaryon formation between different natural isolates is commonly restricted by heterokaryon incompatibility, a common fungal allorecognition mechanism limiting fusion of hyphae to those with the same genetic heterokaryon-compatibility allele combination [16,17]. This severely limits the potential for parasex among natural isolates to generate recombinants, and the interest in researching the role of parasex in nature has declined. The parasexual cycle has mainly been studied and used in laboratory experiments of isogenic strains, e.g., constructing a mitotic genetic map in the asexual *A. niger* [18], for strain construction and complementation of recessive deficiency markers and for strain improvement (e.g., for the over production of glucoamylase [19]), but the natural prevalence and relevance have been largely unexplored.

### 2.3. Sexual Lifecycle

The sexual cycle of *A. fumigatus* was discovered in vitro in 2009 and described the teleomorph *Neosatorya fumigata* [20]. The sexual cycle requires two haploid nuclei of opposing mating-types, MAT1-1 and MAT1-2, which regulate sexual compatibility. After fertilization, a network of dikaryotic cells is formed, which, upon nuclear fusion, produce diploid zygotes, each of which undergoes two meiotic cell divisions, followed by a post-meiotic mitotic division, yielding eight haploid ascospores contained in an ascus (Figure 1) [21]. A fruiting body, termed cleistothecium, may contain several hundreds to thousands of such asci and up to 10^5^ ascospores. A culture may have many cleistothecia, each resulting from a single fertilization process. Growth conditions for sexual replication are very specific, namely at 30 °C on oatmeal agar for 3–6 months in the dark [20]. The natural niche of ascospores has not been detected yet, but they have been hypothesized to exist in composting plant waste material, an environment specifically beneficial for *A. fumigatus* to thrive with high temperatures of up to 60 °C [22]. The contribution of genetic diversity in the life cycle of *A. fumigatus* is difficult to estimate with the currently available knowledge base. The sexual cycle is not as uncommon as may be thought, and it may be the root of the almost infinite variation in *A. fumigatus*, as demonstrated in genotyping studies, but it may be constrained by the environmental conditions under which it is possible.

In the initial study of O’Gorman et al. [20], strains were used that were isolated from air samples in Dublin, Ireland. A few years later the paper of Sugui et al. [23] described the fertility of strains originating from five geographic regions (the United States, Hong Kong, India, England and Ireland), as well as the discovery of highly fertile strains that can complete the sexual cycle in only four weeks, the so-called supermater strains [20,23], showing that, as expected, the fertility of *A. fumigatus* is not restricted to only Irish strains. More recently, a large screening of fertility was undertaken by Swilaiman et al. [24] in a study where a global collection of 131 isolates were screened for the ability to undergo a sexual cycle. Ninety-seven percent of isolates were found to produce cleistothecia with at least one of the available mating partners. Furthermore, interestingly, a large variation was seen in numbers of cleistothecia produced per cross, suggesting differences in the possibility for genetic exchange between strains in nature. Many studies have shown that random sets of *A. fumigatus* strains repeatedly show a 1:1 ratio of the two mating types [22]. This is consistent with the occurrence of sex in nature maintaining a balance of the two mating types, together with evidence of sexual recombination [24]. However, it is clear that it seems highly unlikely to occur within the human body, as opposing mating types are required to initiate a sexual cycle, as well as very specific environmental and nutritional conditions.

## 3. Mutation and Recombination Contribute to Genetic Variation

Genetic variation is created by mutation and recombination and is the source for adaptive evolution. Adaptive variants that arise spontaneously may expand in the population until it either gets fixed or replaced by an even fitter variant that arises. The selection pressures, the frequency of mutation and recombination and the population size all determine the population structure. Although various whole-genome sequencing projects have been undertaken of worldwide sampled *A. fumigatus* isolates as well as clinical samples, the sequencing data has so far not been sufficient to fully understand the population structure. Critical factors that are required to characterize population structure include the extent and nature of standing genetic variation, the contribution by mutation and recombination and what part of the lifecycle is providing this variation and in which environments. Many of these determinants are unclear or partially characterized for *A. fumigatus*. In this section, we explain, for each part of the lifecycle, the potential for providing genetic variation that can contribute to adaptation (e.g., azole resistance) and thereby shape the population structure of *A. fumigatus* next to other known factors that impact population structure, such as environmental pressures.

### 3.1. Asexual Genetic Variation

Asexual reproduction is a common reproductive mode for many fungi, including *A. fumigatus*. With 10⁴ asexual spores per spore head, a colony initiated by the germination of a single spore can easily produce up to 10^9^ asexual spores after one week of growth [12]. Although the number of nuclei needed to populate the mycelium is limited, upon sporulation, the formation of hundreds of millions of spores requires many mitotic divisions and rounds of DNA replication during which replication errors may occur. Despite a very high fidelity with, for example, an estimated 1.67 × 10^−10^ SNP mutation rate per bp per generation for yeast [25], given a genome size of ~30 Mb, millions of the ~10^9^
*A. fumigatus* spores of a three-day old colony are likely to contain a *de novo* mutation in their genome. By chance, some of these mutations may confer resistance to azoles. Due to the high number of new cells generated during sporulation, *de novo* azole-resistance mutations are more likely to occur compared with the expanding mycelium. A large spore production is therefore likely to generate numerous unique genotypes carrying mutations that will be tested for in its current environment by natural selection. In addition, beneficial mutations are more likely to be selected from asexual sporulation due to the single-celled nature of the spores that removes the burden of the (partial) recessivity of mutations that shields the full expression of resistance in a multicellular mycelium [16]. When a beneficial mutation occurs during vegetative growth in the multinucleate mycelium, the resulting resistant nucleus is initially surrounded by wild-type nuclei in a heterokaryotic cell. Mutations may not be fully expressed into the phenotype when wild-type nuclei are also present in the mycelium, and such recessive mutations may segregate and form homokaryotic sectors and sporulate. In addition, upon formation of uninucleate asexual spores, the mutant nucleus can escape from the heterokaryotic mycelium so that, after dispersal and germination, the potentially beneficial phenotype would be fully expressed [26]. The asexual sporulation process thus releases mutations from the mycelium to allow for efficient selection and expression of beneficial traits. Blocking sporulation could thus reduce evolvability. This aligns with the hypothesis that asexual sporulation is essential for both mutation supply and phenotypic expression of azole-resistance mutations in *A. fumigatus* [12]. This has also been shown by Zhang et al. in a study where evolutionary lines exposed to antifungal azole compounds show far less evolvability within mycelium growth and elongation compared to cultures passaged by transferring conidia [12]. In addition to mutation, the asexual cycle may also provide an opportunity for recombination by unequal sister chromatid exchange [27], which may explain the expansion of tandem repeats and azole-resistance development in *A. fumigatus* [28].

### 3.2. Parasexual Genetic Varation

Parasexual reproduction is defined as a ploidy change without meiosis and is often accompanied by mitotic recombination. Parasexuality was first described by Guido Pontecorvo in the 1950s while investigating *A. nidulans* [29,30]. During parasex, fusion of haploid nuclei yields relatively stable diploid cells, which can produce haploid recombinants by mitotic crossing-over and loss of whole chromosomes. Mitotic recombination in contrast to meiotic recombination therefore results in the nonsexual exchange of genetic material. In *A. nidulans*, the potential benefits of a parasexual cycle have been shown by Schoustra et al., who revealed that, by long term passaging of cultures of *A. nidulans*, diploid strains resulted in higher fitness than culturing of corresponding haploid strains [31]. Diploids accumulated recessive deleterious mutations that only became beneficial in recombinant haploid populations. Mutations that are individually neutral or deleterious and that can be beneficial when present in combination are also called sign epistasis. The longer the diploid can evolve, the more mutations it can accumulate and the more it can provide possible genotypes that could potentially enhance fitness. Parasex therefore promotes adaptation or evolution even in the absence of a sexual cycle with its fastidious requirements. A main difference between meiotic recombination and mitotic recombination during parasex is that chromosome dynamics are highly coordinated during meiosis but not in parasex. Coordinated movement of chromosomes has not been reported during parasex, and current models suggest that ploidy reduction occurs by chromosome nondisjunction, leading to an uneven segregation of chromosomes during cell division [21]; for *A. fumigatus,* this has yet to be investigated.

Despite the unknown and possible limited relevance of parasex in nature, recently, it was shown that, in long-lasting fungal colonization of the human lung, heterokaryons can evolve [32]. Here, it was hypothesized that an isolate, initiated from a single spore lasting for many years in the patient, may accumulate somatic mutations during mitotic divisions of the nuclei in the originally homokaryotic culture. Subsequently, heterozygous diploid nuclei may be formed, allowing parasexual recombination. In addition, fusion of isogenic nuclei can yield a homozygous diploid nucleus, which can subsequently evolve into a heterozygous diploid by *de novo* mutations. A heterozygous diploid can continue to accumulate or buffer mutations and finally, after haploidization, segregate into new recombinant genotypes. All these processes may occur coincidentally and contribute to the genetic variability of the culture; the length and likelihood of the sequence of these events is however unknown, and future studies will hopefully provide new insights on this.

Since asexual spores of *A. fumigatus* are uninucleate, newly formed colonies start as a homokaryon and may produce a heterokaryon during mycelial growth due to mutations in some of the nuclei. Heterokaryosis is thus a transient characteristic of the mycelium that is lost upon asexual sporulation and dispersal of spores by air. Therefore, heterokaryons may form and persist, particularly in long-lived mycelium cultures in the human lung (Figure 1). Such is the case for chronic *A. fumigatus* infections or those patients that show colonization and where the fungus has been shown to persist sometimes for many years without active sporulation [16,32]. Evidence for parasexual recombination was found in isolates obtained from CF patients, where *A. fumigatus* is confined to hyphal networks in a biofilm in the epithelial lining of the lung [32].

### 3.3. Sexual Genetic Variation

For *A. fumigatus,* the sexual cycle takes place by the formation of sexual fruiting bodies, or cleistothecia, that, upon completion of the full sexual cycle, can contain up to 10⁴–10^5^ genetically unique ascospores. Haploid progeny analysis revealed extensive genetic recombination for *A. fumigatus* [20]; therefore, the genetic variation that is present in the parental strains is thus enhanced through the reshuffling of these genotypes by recombination. Completion of meiosis is accompanied by formation of recombinant haploid offspring, thus increasing genetic variation upon which natural selection can act. The reshuffling of the genome can, at the same time, also be a disadvantage by breaking up beneficial gene variant combinations (recombination load). Limitation of the sexual cycle therefore appears to be a common strategy for fungi, enabling generation of clonal populations well-adapted to host and environmental niches, yet retaining the ability to engage in sexual or parasexual reproduction to respond to changing environments when necessary [33]. A further limitation may be the heterothallic lifestyle, which requires two strains with an opposite mating type that may not come together frequently. This would explain why the sexual cycle in *A. fumigatus* is still undetected in our environment and could indicate that it is indeed infrequent.

## 4. Antifungal Resistance Selection and the Role of the *A. fumigatus* Lifecycle

The medical triazoles voriconazole, itraconazole, isavuconazole and posaconazole are the most important in the treatment of *Aspergillus* diseases. Triazoles, or more commonly named azoles, are a group of antifungal drugs, and they inhibit the synthesis of ergosterol, which forms a major component of the fungal cytoplasmic membrane [34]. Itraconazole was first introduced for use in patients in 1987, and voriconazole, posaconazole and isavuconazole, as second-generation azole antifungals, were introduced in 2002, 2006 and 2015, respectively. Azole antifungal therapy is recommended for prophylaxis, empiric or pre-emptive therapy for acute disease and long-term maintenance therapy for allergic and chronic pulmonary aspergillosis. Therefore, the emergence of azole resistance threatens the effective treatment of aspergillosis [8]. In the past two decades, azole resistance in *A. fumigatus* has been increasingly reported, both in clinical and environmental strains with tandem repeat mutations in the target of the azoles, the *cyp*51A gene [35,36]. The application of azoles is not limited to clinical use but reaches out to agrochemical applications, as well as use in corrosion inhibitors, dyestuffs and wood preservatives [26]. As azoles are used abundantly in the environment it is thought to have selected for the tandem repeat azole resistance mechanisms in *A. fumigatus* and its global spread [26,37]. Furthermore, the use of similar chemical structures for these various applications has caused cross-resistance between medical and non-medical azoles to develop, also called the environmental route of resistance selection [37]. However, recent studies have challenged the concept of the environmental route, and the importance of the patient route should be reconsidered, since the resistant genotype TR_34_^3^/L98H and TR_120_ both have been recovered from a patient under the long period of clinical azole selection [38,39]. Patient-to-patient transmission has also been investigated, whereby patients can spread *A. fumigatus* into the direct environment by coughing and exposing other patients in the same direct environment [40,41,42]. The same genotype of *A. fumigatus* in cough aerosols and sputum samples were recovered from two out of 15 patients [40]. The assumption that tandem repeat selection via the patient route and patient-to-patient transmission of *A. fumigatus* cannot occur should be reconsidered, even if it accounts for only a minor part.

Clearly, azole resistance is a growing concern, as patients with azole-resistant *A. fumigatus* have a high probability of treatment failure, and alternative treatment options are limited. To obtain a full understanding of azole resistance development, we need to characterize it by what type of mechanisms of resistance emerge, how resistance can spread and if or how resistant genotypes can persist in environments without azoles. The molecular mechanisms that cause antifungal resistance in general are naturally occurring in less susceptible species (intrinsic resistance), or they may be acquired in susceptible strains (acquired resistance). Drug resistance mechanisms can include altered drug-target interactions, reduced cellular drug concentrations mediated by drug efflux transporters or shielding mechanisms, such as biofilm formation [43]. Adaptation is the process by which populations of organisms evolve in such a way to become better suited to their environments as advantageous traits become predominant. Genetic adaptation can be achieved generally by either spontaneous mutation or recombination. As azoles are not known to be mutagenic or recombinogenic per se, they provide a stress factor for *A. fumigatus* and will provoke a strong selection pressure for resistance, as has been shown by many in vitro evolutionary azole resistance selection experiments [12]. In addition to drug resistance, other stress factors (e.g., pH) present in the environment will select adaptive traits.

Adaptation requires genetic variation, which increases the probability that emerging progeny are better suited to survive. As indicated, *A. fumigatus* can benefit from three modes of reproduction to generate variation, but the adaptation potential depends on the availability of these modes in the specific environment, population size and time to complete the life cycle (Table 1). There are some restrictions to each of these modes of reproduction: asexual reproduction requires specific conditions (including humidity, temperature, light, and oxygen); parasexual processes are mostly limited by heterokaryon incompatibility between genetically dissimilar isolates; for sexual reproduction, opposing mating types are required. In human infection, various morphotypes can be observed in different *Aspergillus* diseases. IA is characterized by hyphal growth that causes tissue invasion. The infection is acute and the ability for in-host adaptation seems very limited due to the short duration of disease. In pulmonary cavities, hyphal biofilms may be formed, and asexual sporulation may occur. Cavitary *Aspergillus* diseases are usually chronic, thus creating the potential for high mutation rates. As a consequence, various lineages may develop, including azole-resistant traits [44]. Selection of azole-resistant traits may occur when patients receive antifungal therapy, which may cause the resistant clone to become dominant. Conversion from an azole-susceptible phenotype to azole-resistant phenotype has been repeatedly described in patients with chronic cavities. Genotyping may indicate that the phenotype switch occurred in an isogenic background, supporting in-host selection. However, these observations do not reveal through which reproduction mode the adapted phenotype emerged. Other chronic *Aspergillus* diseases that may provide environments that support adaptation include sinusitis and otitis, although the number of studies that report adaptation remains limited.

Another patient group with chronic *Aspergillus* colonization includes CF-patients. Although lung cavities may develop in CF-patients, *A. fumigatus* is thought to primarily form biofilms in the epithelial mucus. Although this biofilm formation provides shielding from stressors like other microorganisms, antifungal drugs and host immune effectors [45], the fungus cannot benefit from asexual reproduction to adapt to the lung environment, as it is confined to the hyphal morphotype. Nevertheless, antifungal drug resistance reported in azole-treated and azole-naïve CF-patients indicates that *A. fumigatus* employs other strategies that enable adaptation [46]. The study of Engel et al. [32] showed that in chronically colonized patients, *A. fumigatus* can undergo parasexual recombination, characterized by diploid formation. Naturally occurring diploids were previously never detected, but in this study, a large set of 799 *A. fumigatus* isolates were screened that were recovered from contrasting environments, including chronic colonization (i.e., CF and chronic pulmonary aspergillosis), acute IA and from the environment. Diploids were detected in isolates from CF-patients but absent in those from patients with acute infection and the environment. As CF-patients may be chronically colonized with isogenic *A. fumigatus* isolates that form hyphal networks in epithelial mucus, this study showed that the CF-lung might represent a specific niche for parasex to occur. Diploid formation was associated with the accumulation of mutations and variable haploid offspring after crossing-over, including a voriconazole-resistant isolate. Thereby, it was shown that parasexual recombination played a role in azole resistance development in at least one of the detected diploids, which provides a possible explanation for recovery of azole-resistant *A. fumigatus* isolates from azole-naïve CF-patients.

Characteristics of the azole resistance mutations might also provide clues to which reproduction mode was involved. While single resistance mutations in the coding part of the *cyp*51A gene were found in cases of in-host resistance selection, more complex mechanisms involving a combination of SNPs in the coding gene and tandem repeats in the promoter region were observed in resistant isolates, mostly from environmental-resistance selection. Although in-host selection or the patient route of resistance selection may be considered synonymous with asexual reproduction, and environmental resistance selection a combination of sexual and asexual reproduction, over time it has become clear that various resistance mechanisms may develop in both human and non-human environments. A TR_34_^3^/L98H variant was recovered from a CF patient that also harbored a corresponding isogenic TR_34_/L98H isolate, suggestive for a non-sexual genetic alteration. Experimental evolutionary lines that were set up showed that tandem repeat 34 bp promoter elongation is possible via asexual reproduction upon exposure to voriconazole. The TR_34_^3^/L98H first emerged after five cycles, and it subsequently became the dominant genotype with higher azole resistance levels compared to its ancestor strain. This study showed that under strong azole selection, the tandem repeat copy number may increase through asexual reproduction. Mechanistically, this can be explained by replication slippage or unequal sister chromatid recombination during mitosis [37]. Such processes may be rare per single mitotic division but are relevant during growth and sporulation of *A. fumigatus* cultures that involve numerous mitotic divisions. During each of these numerous mitotic divisions, there is a possibility of replication slippage and unequal sister chromatid recombination that may lead to an increase in the tandem repeat number [28]. The observation in the study of Zhang et al. is in line with observations of in-host selection of triazole resistance mutations in a different study, including a 120 bp tandem repeat in the promoter region of the *cyp*51A gene that can be matched with an isogenic ancestor isolate without a tandem repeat from the same patient [38].

There is accumulating evidence that there is an active sexual cycle for *A. fumigatus* that, at least in part, explains the highly genetic diverse population structure in nature [47,48]. As direct observations or sampling of sexual structures in nature have not been reported to date, the implication for sex and azole resistance development have not been elucidated yet. In 2017, a study by Zhang et al. showed evidence for a role of sex in azole resistance selection [22]. Heaps of composting organic waste material were shown to possibly provide the right conditions for sexual reproduction. Next to the commonly detected tandem repeat 46 bp variant, a triple repeat was detected in the samples from this study. An experimental sexual cross that was set up between two tandem repeat 46 bp strains yielded a triple 46 bp variant amongst the progeny isolates. By mispairing in the repeat region of 46 bp during the meiotic process, longer tandem repeats can evolve in the promoter region. In addition to these observations that support sexual reproduction, it is important to note that compost heaps provide favorable conditions for sex, such as a warm, dark, low-O_2_ and high-CO_2_ environment as a result of biological metabolic activity. A dynamic composting process with temperature gradients (20 to 70 °C) and gas changes might therefore stimulate sexual reproduction of *A. fumigatus*. The extent to which these favorable conditions for sex are present in the compost may differ for specific compost samples, which is shown by the differences in growth between the different compost samples after heat shock [22]. A high temperature heat shock is required to induce ascospore germination, and in this study, several samples showed growth after a one hour of heatshock [22]. As composting waste materials contain azole residues from agricultural use, the azole-containing habitat could serve as an evolutionary incubator with increased selective pressure on recombination that benefits the fungus through increased fitness or increased resistance, thus facilitating its survival. This could explain the emergence of tandem repeat 34 bp and 46 bp azole resistance mechanisms that have been found across the globe in a wide range of genetic backgrounds [49,50].

## 5. Final Remarks

Adaptation can be defined as the acquisition of adaptive traits through natural selection, which enables the organism to adjust to live in changing environments. As fungal adaptation results in treatment failure and fungal persistence, the biology of *A. fumigatus* during human infection and colonization needs to be understood to design strategies that can prevent or overcome adaptation. *A. fumigatus* may employ the asexual, parasexual or sexual cycle to adapt to its changing environments. As any change in the environment can provoke adaptation, switching between azoles in patient therapy or in agricultural settings might result in multi-azole-resistant *A. fumigatus* strains through the accumulation of several resistance mutations. When triazole application is stopped, an azole-free environment is created that could prompt selection for compensatory mutations that overcome any fitness costs that are expected to accompany resistance development. As a consequence, there is a risk of selecting for highly resistant strains with wild-type fitness. Future research should investigate the genomic dynamics during infection, as well as in our environment, to understand the key factors facilitating adaptation of *A. fumigatus*.

## Figures and Tables

**Figure 1 jof-07-00599-f001:**
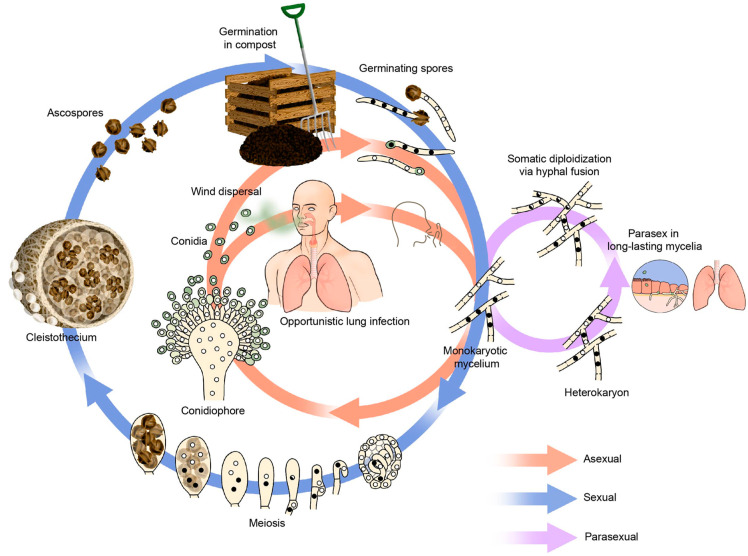
*A. fumigatus* lifecycle and the different ecological niches where (it is assumed) it occurs. *A. fumigatus* can enter an asexual (orange arrow), parasexual (purple arrow) or sexual (blue arrow) reproductive cycle. During the sexual cycle, the mycelium forms a fruiting body, the cleistothecium, which holds the ascospores that, once released into the environment, can colonize ecological niches. Environmental plant waste material, or compost, is an ideal environment for *A. fumigatus* to sporulate, grow, and reproduce, as there are ample resources present, and the moisture and high temperatures inside the compost heap all favor *A. fumigatus*. New mycelium growth colonizes this ecological niche and can enter the asexual cycle. Aerial dispersal of *A. fumigatus* asexual spores from either an environmental niche or from patient-to-patient transmission via coughing (probably rare but should not be excluded) facilitates the spread to patients at risk for opportunistic Aspergillus disease. In contrast to acute IA, *A. fumigatus* can also colonize the lung and form biofilms in patients with structural lung disease, such as CF. The hyphae of an *A. fumigatus* fungal mycelium contain multiple nuclei. Hyphal fusions between compatible monokaryons with genetically different nuclei can yield hyphae with mixed populations of nuclei, called heterokaryons. Subsequently, dissimilar nuclei can fuse to form heterozyogus diploids, which can undergo mitotic recombination by crossing over and haploidization, also known as the parasexual cycle.

**Table 1 jof-07-00599-t001:** Summary of the genetic adaptation characteristics for the different parts of the lifecycle of *A. fumigatus*.

	Asexual	Parasexual	Sexual
**Growth type**	-Unicellular conidia	Multicellular mycelium	Unicellular conidia Multicellular fruiting body Unicellular ascospores
	-Multicellular mycelium		
**Genetic process**	Mitotic mutations and replication slippage or unequal sister chromatid recombination (low frequency 10^−5^)	Mitotic recombination by crossing over and chromosome reassortment	Meiotic recombination by crossing over and chromosome reassortment
**Assumed niche**	-Lung cavity	Long-lasting mycelia in, for example, lung biofilm	Proposed: composting waste material
	-Environmental plant waste substrates		
**Occurrence**	A single spore can form a colony of ~5 cm diameter with up to 10^9^ spores within a week	Diploids form at a rate of ~10^−5^ and haploidize at a rate of ~10^−3^	Sexual reproduction requires mating between isolates of an opposite mating type, ~10^5^–10^6^ ascospores may be formed after ~6 weeks

## Data Availability

Not applicable.

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
