# Peer review of "Azole-Resistance Development; How the Aspergillus fumigatus Lifecycle Defines the Potential for Adaptation"

_jof, 2021, doi:10.3390/jof7080599_

Round 1
Reviewer 1 Report
The review by Zhang et al. aims at discussing how the life cycle of A. fumigatus may explain fungus adaptation. This topic is of interest for the readers of the Journal of Fungi especially to explain the emerging azole resistance.
Comments:
Figure 1, lines 102-103: “Aerial dispersal of A. fumigatus asexual spores from either an environmental niche or from patient-to-patient transmission via coughing”. You might mitigate this statement as evidence for an aerial dispersal of A. fumigatus asexual spores from immunocompromised patients is weak, even if one can acknowledge that the conditions are different in a patient with aspergilloma for example.
Line 185: “may occur, Despite” Please mind the syntax.
Lines 185-186: Can we extrapolate the fidelity occurring in yeasts to A. fumigatus?
Lines 218-219: “Mitotic recombination in constast to meiotic recombination therefore results in the nonsexual exchange of genetic material.” Do you mean “in contrast”?
Lines 253-255: “Evidence for parasexual recombination was found in isolates obtained from CF patients, where A. fumigatus is confined to hyphal networks in a biofilm in the epithelial lining of the lung [28].” Do you mean reference 31 rather?
Lines 368-370: “in cases of proven in host resistance selection” Please rephrase for clarity.
Lines 370-374: “Although in host selection or the patient route may be considered“ Please rephrase for clarity.
Lines 406-409 : “between in compost samples” Please mind the syntax.
Please write A. fumigatus in italics throughout the manuscript.
Reviewer 2 Report
Overall impression:
This is a timely and welcome review as it reminds clinical researchers and others of the importance of knowing your infectious agent. For a long time, the parasexual cycle has been forgotten but for a handful of moulds. Yet, this mechanism of recombination is now gaining importance upon the realisation that the global diversity of Aspergillus fumigatus cannot possibly come about solely through asexual reproduction and in the paucity of evidence for sexuality as a source of variation (although, this reviewer thinks that if you look for something, you will usually find it!). This review addresses this topic to some extent.
However, there are significant but not insurmountable amendments that need to be made before it can be published. As a review, it should state all sides of the argument, different viewpoints, and place facts in context according to the main aim (antifungal resistance) as well as offer opinions where warranted. This reviewer’s opinion is that the breadth of considerations needs to incorporate other life cycle (lifestyle?) factors that are important to the adaptability of A. fumigatus for the development of resistance. As an example, on p. 5, lines 186-188, in the sentence “…given a genome size of ~30Mb, millions of the A. fumigatus spores of a colony are likely to contain a de novo mutation in their genome”, is implied to be the only mechanism for the genetic variation observed globally. It seems that this section does not seem to represent the various nuances of asexual reproduction, that actually, A. fumigatus is much more plastic in its adaptability and doesn’t necessarily require mutations to adapt. As evidence for this complexity, a recent paper describes evidence that genetically identical individuals show phenotypic variation that is increased by environmental stress (https://www.biorxiv.org/content/10.1101/797076v1.full) and provides another viewpoint for the role of asexual reproduction/lifestyle. A review should present all the facts but then dissect, discuss and even contrast them. This is lacking to some extent at the start of the review, although section 4 offers this essential component. In addition, there are other Aspergilli whose parasexual cycles are better studied, which should be mentioned in context.
The review also suffers from lack of attention – in places it reads as if it is the first draft, although it is very well-written in others (section 4 particularly, where the essence of the aim of the review is discussed). The review could be improved immensely if it was reread critically and revised with attention to detail with regards to spelling, italics and word spacing. It is also repetitive in some areas. It could be condensed substantially with some critical editing. This reviewer suggests that the first section – “2. A. fumigatus lifecycle” is too general for some sections (e.g., asexual lifecycle”) and includes material that should be in section 3 “Mutational and recombinational supply to genetic variation” anyway (e.g., parasexual life cycle). The most concise option would be to combine these sections and edit significantly to remove repetitiveness and awkward phrases.
Notwithstanding these comments, I am wholeheartedly in favour of publishing this review as it is of importance to scientific and clinical researchers.
Detailed comments: those in bold require particular attention
It is debatable whether the title should comprise a subordinate clause. This reviewer suggests it could be rephrased to “The plasticity of the Aspergillus fumigatus lifecycle defines its adaptive potential for developing azole-resistance;” without losing its impact.
p2, line 46: please change “capable to persist” to “capable of persisting”
p2, line 59: please italicise “A. fumigatus” in the heading. “A. fumigatus”, “Aspergillus”, “A. nidulans”, “de novo”, “in vitro”, “Neosatorya fumigata” all require italics. Please check the manuscript and correct throughout.
p2, line 61: correct the spelling of “hyphae”, presumably meant to be singular (hypha)
p.2, line 65: “sporehead” is two words, please correct
p2, Figure 1: suggest that “Diploidization via hyphal fusion” be changed to “Somatic diploidization via hyphal fusion” as the label of the parasexual phase for clarity and to differentiate this process from the diploidization during meiosis
p3, line 99 (in the legend of Figure 1): please correct the spelling of “plante”
p3, line 101 (in the legend of Figure 1): please change “the ecological niche and can re-enter” to “this ecological niche and can enter” to improve clarity
p3, line 101 (in the legend of Figure 1): please correct the spelling of “asxual”
p.3, line 102-103 (in the legend of Figure 1): “patient-to-patient transmission via coughing facilitates the spread to patients at risk for Aspergillus disease”. This form of transmission is very rare (three reports (see below), including one from some of the current authors) and the phrase “facilitates the spread to patients” should be tempered to reflect this rarity; also, “opportunistic” should prefix “Aspergillus disease”.
https://wwwnc.cdc.gov/eid/article/25/4/18-1110_article
https://wwwnc.cdc.gov/eid/article/24/8/17-1865_article
https://academic.oup.com/cid/article/34/3/412/390453
p. 3, lines 104-107 (in the legend of Figure 1): this last part of the legend requires some reworking
to improve clarity. This reviewer’s suggestion is below.
“The hyphae of an A. fumigatus fungal mycelium contain multiple nuclei. Hyphal fusions between compatible monokaryons with genetically different nuclei can yield hyphae with mixed populations of nuclei, called heterokaryons. Subsequently, these nuclei can fuse to form heterozyogus diploids, that can undergo mitotic recombination and haploidization, also known as the parasexual cycle.”
p.3, line 110: the importance of “sex in nature” should be explained here, i.e., to create diversity.
p3, line 112: suggest the revision of “…genetically different colonies…” to “vegetatively compatible
but genetically different colonies”
p3, line 116: delete the word “by”, not needed.
p3, line 122-123: delete the “only” in “…limiting fusion of hyphae only to those with the same genetic heterokaryon-compatibility allele combination.”
p3, line 125: suggest amending “…the interest in the role of parasex…” to “the interest in researching the role of parasex…” for clarity
p. 3, lines 128-129: suggest changing “…remained doubtful.” to “…are largely unexplored.”
This reviewer suggests that there is a lack of published work in this review with regards to parasexuality in the context of its industrial use, which could be cited in this subsection of the review (as it is for the sexual life cycle section that follows it).
p4, line 132: please amend “…MAT1-1 and MAT1-2.” to “…MAT1-1 and MAT1-2, which regulate
sexual compatibility” for clarity
p.4. lines 133-135: the sentence “After fertilization the formation of a network of many dikaryotic cells that upon nuclear fusion produce many diploid zygotes that each undergo the two meiotic cell divisions followed by a post-meiotic division yielding eight haploid ascospores contained in an ascus (Figure 1).” is awkward. This reviewer’s suggestion to amend this sentence is below.
“After fertilization, a network of dikaryotic cells is formed, that upon nuclear fusion produce diploid zygotes, each of which undergo two meiotic cell divisions followed by a post-meiotic mitotic division yielding eight haploid ascospores contained in an ascus (Figure 1).”
p.4, line 139: please insert a comma after “specific” for clarity
p.4, line 144: please replace “…current available knowledge.” with “currently available knowledge base.” or “current knowledge base.”
p4, lines 146-147: please replace “requirements” with “conditions” to avoid repetition
p4, line 155: replace “screening” with “screened”
p.4, line 163: Section “3. Mutational and recombinational supply to genetic variation” title – replace
“supply” with “contribute”
This reviewer suggests adding a concluding remark in the section “Sexual lifecycle” that the sexual cycle is not as uncommon as may be thought, and may be the root of the almost infinite variation in A. fumigatus as demonstrated in genotyping studies, but that it may be constrained to the environment by the conditions under which it is possible. In fact, the sentence “However, it is clear that it… and nutritional requirements.” (p. 4, lines 144-147) should be moved to the end of the second paragraph in this section and the contrast between sex in environmental and clinical strains be made for more impact.
p.4, line 169-170: correct the spelling of “sampels”
p4, lines 163-177: Comment on “3. Mutational and recombinational supply to genetic variation”: This reviewer argues that although genome sequencing studies are required to understand population structures, there is something to be said for understanding the role of the different parts of the life cycle by performing good genetic studies on well-chosen individual strains.
p4, line 175: please replace “In this paragraph…” with “In this section…”
The authors could improve the review significantly by focusing on the contrasting pressures on A. fumigatus in different environments, such as nature (against fungicides) and in the human body (against therapy), where the fungus is constrained to asexual and parasexual life cycles (that is, as far as we know for now). (note added post-reading: the reviewer acknowledges that this topic is discussed in the last section of the review but this line of thought can be the thread that links the different sections together)
p5, lines 180-181: that a colony only produces 109 spores from germination of a single spore seems
low, certainly for one week’s worth of growth – is this in vitro or in the wild?
p.5, lines 176-177: please rephrase “…and thereby also shape the population structure of A. fumigatus.” A population structure and its resistance patterns are dependent on many more factors that just the life cycle of A. fumigatus., e.g., environmental pressures.
p.5, line 180: “sporehead” is two words, please correct
p5, line 183: please delete “on top of the mycelium”, this phrase is awkward, incorrect and unnecessary
p5, lines 183-184: replace “will require” with “requires”
p5, line 185: a full stop is required after “errors may occur” for clarity
p5, line 185: in the sentence, “Despite a very high fidelity…”, please insert “in replication” for
clarity. This whole sentence needs commas to improve clarity.
p. 5, line 218: “loss of whole chromosomes” – this is an important concept in the adaptability of A. fumigatus to the host and is not, but should be, mentioned if not discussed as an alternative theory in this review.
Addition of a detailed figure in the section illustrating the chromosomal mechanisms contrasting asexual and parasexual cycles in section 3 would be very useful and not out of line for a review.
p6, lines 233-234: in “…during cell division [20], for A. fumigatus this has yet to be investigated.” change the comma to a semi-colon.
p6, lines 244-245: for the statement “All these processes may co-occur and contribute to the genetic variability of the culture.” please provide the likelihood of all the options discussed in the preceding paragraph to provide context.
p6, line 246: please replace “co-occur” with “occur coincidentally”
p6, lines 246-247: explain how a heterkaryon is produced from a homo/monokaryon in the phrase “…newly formed colonies start as a homokaryon and may produce a heterokaryon during mycelial growth”
p6, line 249: please explain that “heterokaryons may form and persist” only if the mycelia are vegetatively compatible; note that this is not likely unless a mutation occurs in the parent strain that yields heterokaryotic hyphae OR another compatible mycelium present in the lung anastomoses with the parent hypha to establish a heterokaryon, this negating the idea of single spore origin of this heterokaryon.
p6, lines 262-263: please temper the statement “sporulation of the newly formed recombinant haploid offspring” with the fact that spore survival rates are probably low.
p6, line 267: “well adapted” needs a hyphen
p6, lines 263-264: “…upon which natural selection can act.” is used to describe selection pressures following sexual reproduction but surely, these pressures also function in the other 2 parts of the cycle?
p.7, line 285: delete the first “gene” in the sentence “…in the target gene of the azoles, the cyp51A
gene.” to avoid repetition.
p7, line 291: please provide a reference for the statement “has caused cross-resistance between medical and non-medical azoles to develop”
p7, line 291: please replace “the” in “for the various applications” with “these”.
p.7, lines 295-299: For the sentence “Patient-to-patient transmission has also been investigated whereby patients can spread A. fumigatus into the direct environment by coughing and exposing other patients in the same direct environment. The same genotype of A. fumigatus in cough aerosols and sputum samples were recovered from two out of 15 patients [39].” As this review has only provided one reference to evidence this phenomenon (and this reviewer has provided only 2 more, see above), it seems that patient-to-patient transmission is rare. Please add this qualifier. Moreover, there is published evidence for the occurrence of TR34 or L98H on their own found in CPA patients exhibiting azole resistance, which would add credence to this theory.
p7, line 308: the phrase “…in strains of susceptible organisms.” is awkward; please rephrase to “…in susceptible strains.”
p7, line 316: delete “azole” in “…azole resistance” as it is redundant in this sentence.
p7, line 318: can you provide a refence(s) or more detail to support “…other stress factors present in the environment will select traits…”
p7, lines 320-321: The sentence “Adaptation requires genetic variation, as the probability of progeny emerging that is better suited to survive increases.” is awkward. This reviewer suggests the following alternative:
“Adaptation requires genetic variation, which increases the probability that emerging progeny are better suited to survive.”
p7, lines 321-322: for clarity and to improve flow, please replace “…three reproduction modes…” with “…three modes of reproduction…”
p7, line 322: please replace “will depend of” with “depends on”
p7, line 325: please correct the spelling of “oxgeny”
p7, lines 324-325: the authors state that “While there are no apparent restrictions for A. fumigatus to undergo asexual sporulation (besides humidity, temperature and light, oxgeny [sic])…” – these 4 factors sound like a lot of restrictions! Please temper this statement. For example, although there are some restrictions on asexual reproduction (including humidity, temperature, light, and oxygen), there is an additional limitation of requiring a vegetatively compatible to achieve heterokaryosis during parasexuality, and …etc.
p7, lines 329-330: please cite the evidence for “…the ability for in-host adaptation seems very limited
p7, lines 330-331: “…hyphal biofilms and asexual sporulation may be formed.” is awkwardly worded. Please replace with “…hyphal biofilms may be formed and asexual sporulation may occur.”
p8, line 332: please replace “…high mutational supply.” with “high mutation rates.” to improve flow.
It seems that at the end of section “4. Antifungal resistance selection and the role of the A. fumigatus lifecycle” how mutations derived through the parasexual cycle can actually yield a
resistance mutation, like TR34/L98H for example, have not been explained. This reviewer was looking forward to having this mechanism detailed more completely, although the discussion of the mechanism of acquisition of the TR34 mutation on p. 9, lines 367-390 goes some way to providing this.
p8, lines 335-338: For the statements “Conversion from azole-susceptible phenotype to azole- resistant phenotype has been repeatedly described in patients with chronic cavities. Genotyping may indicate that the phenotype switch occurred in isogenic background supporting in host selection.” this is only one side of the story. In many patients with long-term/chronic aspergillosis, there is no reason to believe that they are not colonised with (many) different strains that become more/less dominant during successive exacerbations. There are many reports of this evidence as well, which needs to be mentioned.
p8, lines 344-366: this section detailing the CF example is excellent. There is one point to address and that is the phrase “Diploids were prevalent in isolates from CF-patients,…”. The Engel study found that diploid strains were recovered from “six of 11 (55%) CF patients” – is that prevalent? Please clarify.
p.9, line 369: please define “TR”
p9, line 377: please rephrase “…when exposed to voriconazole” to “...upon exposure to voriconazole” to improve flow.
p9, line 392: please correct the expression “at least for a part” to “at least in part”
p9, line 399: please replace “experimentally” with “experimental”. Moreover, the Zhang sexual crosses could be illustrated in a figure to relay this information visually (and arguably, more clearly).
p9, line 403: please replace “…such as warm,…” with “…such as a warm,…” to improve flow.
